# Fire Egress System Optimization of High-Rise Teaching Building Based on Simulation and Machine Learning

**Muchen Zhou, Bailing Zhou *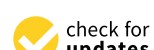, Zhuo Zhang, Zuoyao Zhou, Jing Liu, Boyu Li, Dong Wang and Tao Wu**

School of Urban Construction, Wuhan University of Science and Technology, Wuhan 430065, China; wutao@wust.edu.cn (T.W.)
* Correspondence: zhoubailing@wust.edu.cn

**Abstract:** A fire egress system is one of the most critical aspects of fire emergency evacuation, which is the cornerstone technology of building fire safety. The high-rise teaching buildings on campus, where vast crowds of people gather, need to be qualified for rapid evacuation in the event of a fire especially. Conventional teaching building egress system design places more emphasis on individual elements (e.g., stairwells, evacuation doors, and evacuation walkways) rather than on their co-regulation as a whole. Furthermore, there are not enough holistic and effective optimal design strategies, which is because most of the existing studies rely on experiments or simulations and often suffer from a lack of sufficient data to fully reveal the interactions of individual variables. In this study, the co-effectiveness of stairwells, walkways, and room doors in reducing total evacuation time was investigated by simulation and machine learning. We selected a typical high-rise teaching building as an example and integrated two simulation software, Pyrosim and Pathfinder, to compare the available safe evacuation time (ASET) and required safe evacuation time (RSET). Then, a framework consisting of five factors—stair flight width (SFW), stairwell door width (SDW), corridor width (CW), room door width (RDW), and location of the downward stair flight (LDSF)—was established for the optimization through statistical analysis of big data obtained by the preferred machine learning algorithm. Results indicate that (1) By modifying just one factor (SFW), the total evacuation time (TET) can be reduced by at most 12.1%, with the mortality rate dropping from 26.5% to 9.5%; (2) although ASET could not be achieved either, among 4000 cases of multi-factor combinations, a maximum TET improvement degree, 29.5%, can be achieved for the evacuation optimization compared to baseline model, with a consequent reduction in mortality to 0.15%; (3) it shows that the emphasis of the egress system optimization is on the geometric features of the evacuation stairwell; furthermore, the multi-factor combination approaches have better compromised evacuation performances than the single-factor controlled schemes. The research results can be applied as rational design strategies to mitigate fire evacuation issues in high-rise teaching buildings and, in addition, the methodology suggested in this paper would be suitable to other building types.

**Keywords:** fire safety; egress system optimization; simulation; machine learning



## 1. Introduction

### 1.1. Background

While fire has brought civilization to mankind, it has also brought disaster to us, leading to the loss of property and even human life [1]. There are around seven million fires globally every year [2], including nearly 14,500 in high-rise buildings [3]. With the advancement of construction technology, high-rise buildings have become very popular in the world, and high-rise teaching buildings are one of the common types. A high-rise teaching building on university campuses generally consists of classrooms, offices, studios, laboratories, etc., which is usually a typically staff-intensive place. According to statistics [4], more than 4000 fires have occurred in universities across China over the past decade, mostly in classrooms and laboratories. These catastrophes resulted in the deaths of

over 50 students and educators, with total economic losses amounting to USD 25 billion. Hence, the issue of fire safety in high-rise teaching buildings has received considerable critical attention.

The core purpose of fire safety is to protect human life. The most fundamental and important technology for the fire protection design of buildings is, specifically, fire emergency evacuation. Ensuring occupant safety in fire hazard involves a number of essential aspects, including behavioral patterns of escapees, fire development process, building structure, building fire prevention facility configuration, and so on. The purpose of fire emergency evacuation is to make sure that the RSET is less than ASET.

As an important part of fire safety, the egress system made up of a variety of components, including stairs, refuge floors, sky bridges, etc., has a significant impact on the comprehensive evacuation performance of the building. Additionally, the geometric features of the egress system elements, such as the number and placement of staircases, influence how well it works [5]. These variables are defined in the stage of architectural scheme design, and once the plan is put into place, it cannot be changed afterward. Therefore, it is critical to select suitable parameters at the outset of the schematic design phase, which will ensure the effective operation of the egress system.

Based on the current study, the principal objective of this paper is to reveal the intrinsic linkages and combined impacts of egress system factors in high-rise teaching buildings by means of simulation, machine learning, and statistical analysis, which provides a basis for choosing parameters during the architecture scheme stage. Furthermore, this information would serve as a theoretical and technical foundation for development and remediation in the field of building egress system optimization.

*1.2. Literature Review*

Building egress systems have been extensively studied, most of which were conducted in densely populated places such as hospitals [6,7], campuses [8–10], subway stations [11–13], and airports [14].

One of the most important advances in this field was that Tweedie et al. [15] first calculated evacuation durations by using a computer simulation of the fire evacuation procedure in the 1980s. Since experimenting with fire in the real world would result in accidents or severe injuries, the simulation method marked an important turning point for the research of fire evacuation [11]. Recent years have witnessed a growing academic interest in the areas of fire lighting simulation [16], fire smoke flow simulation [17], fire safety assessment [18,19], fire evacuation simulation [20–23], etc.

The rapid development of computer computing power allowed for more detailed, complex scenarios of three-dimensional fire simulations, the use of which can facilitate research on this topic [24]. While computer simulations have undoubtedly made the study more convenient, what is known about fire evacuation comes from small-scale experiments with inadequate samples [25]. Given that data analysis is a key method of the scientific research [26], small sample sizes have been a serious limitation for previous studies on fire emergency evacuation.

Recent advances in machine learning have made it easier to expand the data capacity and build the sample database [27]. Because of its advantage of extracting hidden rules from real data, it has a broad range of developments in computer science and is quickly expanding to economics and medicine [28], and then to a wide range of fields. Up to now, several studies have been developed around the theme of machine learning, such as investigating the factors influencing people movement patterns during evacuation [29], detecting the trend of crowd flow during evacuation [30], creating a training system to improve evacuation capability by inducing crowd movement state through dynamic guidance signs [31], applying ANN to precisely model people's behavior during evacuation and their responses to other people and obstacles [32], and developing a rescue route planning algorithm that takes credit for all aspects of local safety performance [33]. These

papers demonstrate how useful the prediction results of machine learning can be as a basis for evacuation studies.

The main focus of the studies in building egress systems included exit numbers [9], exit widths and locations [34], stairway forms [8], stair number and locations, stair widths [5], and corridor widths [34,35] and lengths [34]. In addition, most of them only discussed a specific stage of the evacuation process, such as the evacuation from the room to the RD [36], the evacuation from the RD to the walkway [37], and the evacuation in the stairwell [5,38]. However, few writers have been able to draw on any structured research into the entire building evacuation process. Moreover, there is a lack of analytical studies on the combined effects of various design variables.

The summary of literature is shown in Table 1.

**Table 1.** Summary of the research.

| References | Research Tools | Scenarios | Content |
|---|---|---|---|
| Fang, Z.M. (2012) [38] | Empirical study | High-rise commercial building | Factors affecting the evacuation speed of stairwells |
| Li, Y. (2020) [6] | Simulation | Hospital | Effect of overlap and acceleration on evacuation |
| Zang, Y. (2021) [10] | | Campus | Effect of obstacles on evacuation |
| Zhang, X. (2018) [23] | | Subway station | Effect of floor plan on evacuation |
| Liu, Y.Q. (2021) [17] | | Subway station | Fire smoke flow simulation |
| Wang, N. (2021) [19] | | Underground shopping malls | Fire safety assessment |
| Tajima, Y. (2001) [36] | | | Effect of door size of exit on evacuation |
| Weifeng, F. (2003) [37] | | | Bidirectional pedestrian movement characteristics |
| Rostami, R. (2021) [9] | | Elementary school | Effect of parameters such as exit numbers on evacuation |
| Kodur, V.K.R. (2020) [5] | | High-rise office building | Effect of stair location on evacuation |
| Li, J.C. (2022) [34] | | | Optimal ratio of parameters for convex exit |
| Syed Abdul Rahman, S.A.F. (2021) [35] | | Campus | Evacuation emergency management |
| Wang, K. (2019) [29] | Machine learning | | Evacuees' movement pattern |
| Horii, H. (2020) [30] | | | Identification of crowd behavior |
| Gu, J.L. (2022) [31] | | Campus | Emergency management and evacuation simulation |
| Tkachuk, K. (2018) [32] | | | Prediction of the evacuation route |
| Wang, K. (2023) [21] | Intelligent algorithm | | Evacuation route optimization |
| Deng, H. (2021) [33] | Simulation, machine learning | Campus | Evacuation route planning |
| Guo, K. (2022) [11] | | Subway station | Evacuation optimization |
| Zhong, Y. (2021) [16] | Simulation, intelligent algorithm | | Fire emergency lighting distribution |

### 1.3. Scientific Originality

Fire emergency evacuation includes numerous processes that each integrate a wide variety of components, while component characteristics determine the building egress system. Previous studies have predominantly focused on one particular stage of evacuation or investigated how one particular component affects evacuation, which ignored the linkage effect between multiple components. Moreover, a lot of earlier studies only used data from simulation, which were somewhat limited due to their lack of sample sizes.

To this end, we (1) generalized all the egress system components and their parameters, allowing the optimization gaze to be more thorough and multifaceted; (2) counted on machine learning to gather vast amounts of basic research data and compare them to the evacuation performance of baseline model in order to investigate the potential for optimization; (3) utilized Design Explorer software to visualize the data analysis in order to examine the interaction effects of the variables and generate suggestions for improving evacuation performance. (4) Additionally, because different types of buildings have similar egress system components, the workflow put forth in this paper can be used to optimize evacuation of other building types.

### 1.4. Aim of This Work

This paper aimed to address the following research questions: (1) How important is each factor and how much does it affect the building egress system when several independent factors are in play at once? (2) Is it possible to identify a collection of solutions that work best for each variable parameter to reduce TET? To achieve the research objectives, this study used a real-world building case to test the efficacy of the analysis method, and it created a database using a preferred machine learning algorithm to enhance the single simulation method and lower the likelihood of the outcomes. Our major contributions were: (1) Several variables in the evacuation process route were taken into consideration, and sensitivity analysis was used to determine the contribution potential and weights of each factor to the evacuation efficiency; (2) the interactive combination effects of the variables were investigated through data screening and comparison; (3) the study results would offer a foundation for parameter choice in architecture scheme design and enrich the research framework of building egress system optimization.

The rest of the essay is laid out as follows: a four-step framework of high-rise teaching building egress optimization is established in Section 2. In Section 3, the outcomes of single-factor effects and multi-factor combination effects based on simulation and preferred machine learning algorithms are displayed. Finally, Section 4 concludes the research.

## 2. Research Methodology

In this work, Pyrosim simulation software was used to determine the ASET, Pathfinder simulation software was utilized to simulate the fire evacuation process for obtain the RSET. Then, closely followed by MATLAB (R2019b Update9) software to preprocess, manipulate and sample the data, machine learning algorithms were continuously introduced to effectively study large-capacity samples, and ultimately suggest evacuation optimization strategies through a series of data analysis. The following is the technology workflow for this research (Figure 1).

### 2.1. Study Building Specification

This paper only centered on the second teaching area of the building, which has ten floors and is located in the middle of the building's three distinct teaching areas (Figure 2a). Each level in this area is 4.2 m high, with a total building height of 52.5 m (Figure 2b). Its symmetrical layout, as seen in Figure 2c, features the same general layout on each level with a total of four staircases. The atrium is only present in the center of the first through seventh stories, and a corridor has also been created in between the two stairs on the east side of the fourth and fifth levels. Auxiliary rooms and restrooms are also included on the west side of the floor layout.

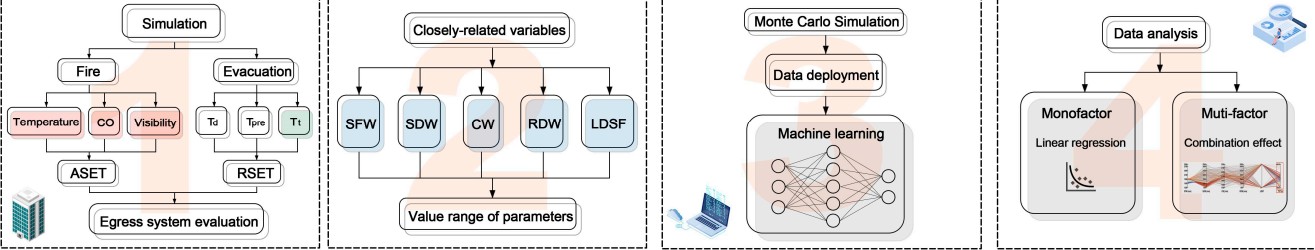

**Figure 1.** Research framework.

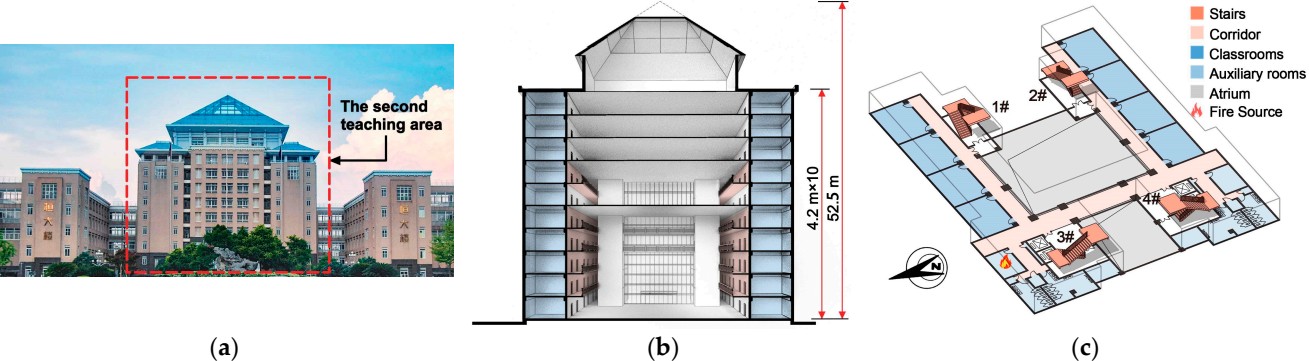

| (a) | (b) | (c) |
|---|---|---|

**Figure 2.** (**a**) Photo of the teaching building; (**b**) model of the teaching building; (**c**) typical plan and fire source setting.

The main cause of fire fatalities is smoke from combustion [39], and due to the chimney effect, smoke will quickly travel to higher floors through vertical traffic nuclei such as stairwells and elevator shafts, decreasing visibility and producing hazardous gases to impede evacuation progress. Therefore, when the fire source is situated on a lower floor, the impact is greater and more widespread [5].

Given the real circumstances of the case, the fire source was set in the second-floor substation room, which is adjacent to the storage room (Figure 2c). The fire is thought to have been sparked by an old electrical wire short circuit, and the textile, plastic, and paper products in the warehouse helped it spread more quickly. The scenario setting essentially replicated the fire's unfavorable circumstances.

This research focused on the design parameters of evacuation-related components in buildings, excluding the effect of positive fire protection systems. Elevators are not included in the essential egress coordination, and a number of extra conditions must be fulfilled before they can be used in a fire emergency [40]. Furthermore, many nations expressly forbid using elevators as a means of evacuating citizens in the event of a fire [41]. As a result, it is predetermined that an emergency cannot be handled by using the elevator.

### 2.2. Indicators for Evaluations of the Building Egress System

By comparing the available and required safe evacuation time, it is possible to assess the building egress system. The building egress system complies with the evacuation requirements if the ASET is greater than the RSET, and vice versa [42] (Table 2).

**Table 2.** Building egress system evaluation.

| Scenarios | Evaluation |
|---|---|
| ASET > RSET | Safe |
| ASET < RSET | Dangerous |

2.2.1. Criterion for Evaluation of the ASET

In this paper, Pyrosim 2021 software was chosen to simulate the fire in order to determine the ASET. Developed by the National Institute of Standards and Technology (NIST), Pyrosim software is particularly intended for fire dynamics simulation (FDS) [43]. By setting parameters such as material combustion performance, fire source location, and combustion release rate, it can effectively portray the smoke development situation under actual fire scenarios [44].

During the fire simulation trials, the grid parameters needed to be set to ensure the accuracy of the computation. Typically, the size of the mesh cell ($\delta x$) is linked to the diameter size of the fire characteristic ($D^*$) with a ratio ($D^*/\delta x$) between 4 and 16. The following formula can be used to calculate the diameter size of the fire characteristic ($D^*$) [45]:

$$D^* = \left( \frac{Q^*}{\rho_\infty c_P T_\infty \sqrt{g}} \right)^{\frac{2}{5}} \tag{1}$$

where $Q^*$ is the overall heat release rate of the fire (KW), $\rho_\infty$ is the air density (1.204 kg/m$^3$), $c_P$ is the air specific heat (kJ/(kg·K)), $T_\infty$ is the ambient temperature, and g is the acceleration of gravity (m/s$^2$).

Given that the assumed ambient temperature was 20 °C, the grid was speculated to be between 0.23 and 0.93 m [46]. In view of the software running duration and model calculation accuracy needs, the grid size was set to 0.25 m × 0.25 m × 0.25 m [47], making simulation results sensitive to the mesh employed (554,812 grids). The overall simulation duration was 600 s [43], the fire source area was set to 1 m × 1 m, with fire heat release rate set to 6 MV [48] and the combustion reaction set to polyurethane combustion [49]. The fire growth time can be calculated as follows [46]:

$$Q = \alpha t^2 \tag{2}$$

where Q is the fire heat release rate (KW), $\alpha$ is the fire growth coefficient (KW/s$^2$), t is the time (s). The fast-growing fire mode, with a fire growth coefficient of 0.0469 KW/s$^2$, is selected based on the case's real circumstances. It is clear from the computation that the fire will burn for 361 s before reaching its maximum value of the chosen heat release rate. Hence, the fire simulation duration was chosen to be 600 s in order to achieve more complete simulation data [43].

In general, the time it takes for temperature, CO concentration, or visibility to reach a critical level is used as an indicator for ASET. According to previous literature, people will become exceedingly uncomfortable and unable to breathe when the temperature reaches 60 °C [50]; hence 60 °C is utilized as the critical danger temperature. People will asphyxiate quickly when the CO concentration hits 500 ppm. Therefore, the hazard threshold for CO is 500 ppm [50,51]. Although the Australian Guide for Fire Engineers states that the 5 m visibility level is the standard for small places [52], previous study [53] unequivocally discovered that walking speed decreases sharply when visibility reaches 1 m, and then stops changing relatively significantly even visibility continues to decline. Thus, the moment when visibility reaches 1 m is identified as one of the criteria for determining the value of ASET. The mentioned data above are detailed in Tables 3–5.

**Table 3.** Critical danger temperature setting.

| Temperature (°C) | Endurance Time (min) |
| --- | --- |
| <60 | >30 |
| 100 | 12 |
| 180 | 1 |

**Table 4.** Critical danger CO content setting.

| CO Content (ppm) | Exposure Time | Harm Effect |
|---|---|---|
| 100 | Within 8 h | No feeling |
| 400–500 | Within 1 h | No feeling |
| 600–700 | Within 1 h | Headache, nausea, breathing disorder |
| 1000–2000 | Within 2 h | Consciousness, breathing disorders, coma, die within 2 h |
| 3000–5000 | Within 20–30 min | Death |
| 10,000 | Within 1 min | Death |

**Table 5.** Critical danger visibility setting.

| Visibility Threshold (m) | Scenarios |
|---|---|
| 1 | Small spaces |
| 10 | Large spaces |

2.2.2. Criterion for Evaluation of the RSET

In this paper, Pathfinder software was chosen to simulate evacuation to obtain the RSET. It can be calculated as follows [54]:

$$RSET = T_d + T_{pre} + T_t \tag{3}$$

where $T_d$ signifies the fire detection alarm time (s), $T_{pre}$ denotes the personnel pre-movement time (s), and $T_t$ represents the personnel evacuation movement time (s).

Since this paper aimed to investigate the influence of building design parameters on evacuation, excluding $T_d$ and $T_{pre}$ from consideration, the values of both were assumed to fall within a fixed range [41] (Figure 3).

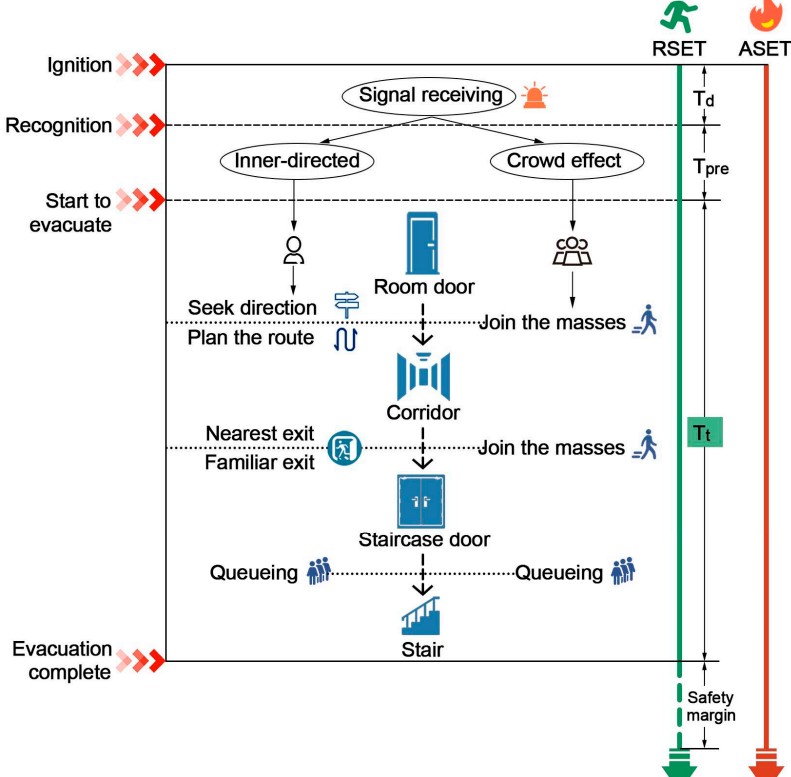

**Figure 3.** ASET and RSET.

There are two operating modes in Pathfinder software: SFPE and Steering. Steering mode includes complex behavioral aspects of evacuation process, such as collision avoidance and route selection, which can accurately simulate the evacuation process of people by setting characteristic parameters (height, gender, shoulder width) and motion parameters (walking speed) [5]. The formula for the motion scenario is shown in detail as follows [10]:

1.  The speed of evacuating people decelerates when passing through obstructions such as stairs, with a certain acceleration maintained in areas with low density of passenger flow:

$$v_0 = \begin{cases} v_{max}\frac{k}{1.4}(D < 0.55) \\ v_{max}\frac{k-0.266kD}{1.19}(D \geq 0.55) \end{cases} \tag{4}$$

$$a_{max} = \frac{v_{max}}{t} \tag{5}$$

where $v_0$ is the evacuees' starting velocity (m/s), $v_{max}$ is the evacuees' maximum velocity (m/s), k is the impact factor, D is the density of evacuees', (per/m$^2$), $a_{max}$ is the maximum acceleration (m/s$^2$), and t is the acceleration time (s).

2.  Directional selection weights during evacuation:

$$\omega = \frac{\theta}{2\pi} \tag{6}$$

where $\omega$ represents the significance of the chosen evacuation direction, and $\theta$ represents the angle between all possible evacuation directions and the curve tangent of the software-planned path.

3.  The speed and acceleration vectors in the direction of evacuation with the least probability of path selection:

$$\left|\vec{v}\right| = \begin{cases} 0 \ (l_{max} \leq l_{stop}) \\ v \ (l_{max} > l_{stop}) \end{cases} \tag{7}$$

$$\vec{v}_{min} = \left|\vec{v}\right| * \vec{l}_{min} \tag{8}$$

$$\vec{a}_{min} = \frac{\vec{v}_{min} - \vec{v}}{\left|\vec{v}_{min} - \vec{v}\right|}a_{max} \tag{9}$$

where $\vec{v}$ is the speed of vector in the present evacuation direction (m/s), $l_{max}$ is the longest forward distance in the present evacuation direction (m), $l_{stop}$ is the smallest acceleration-affected distance in the present evacuation direction (m), $\vec{l}_{min}$ is the least important evacuation direction, $\vec{v}_{min}$ is the least valued velocity vector in the evacuation direction (m/s), and $\vec{a}_{min}$ is the least influenced acceleration vector in the evacuation direction (m/s$^2$).

4.  The speed and place of evacuees travelling to the next location:

$$\vec{v}_{next} = \vec{v}_{min} + \vec{a}_{min}\Delta t \tag{10}$$

$$\vec{P}_{next} = \vec{P} + \vec{v}_{min}\Delta t \tag{11}$$

where $\vec{v}_{next}$ is the speed of the evacuees when they reach the next location, $\vec{P}_{next}$ is the next location, $\vec{P}$ is the present location, and $\Delta t$ is the duration.

### 2.3. Determination of Main Independent Variables

With regard to the literature currently in print, the primary variables for the evacuation process are stair width [5], corridor width [34,35], and door width [55]. In this paper, by classifying the process routes of evacuation, we synthesized the aforementioned variables and list five pertinent factors: RDW and CW play a role in the horizontal evacuation process, while SDW, SFW, and LDSF play a role in the vertical evacuation process, where "LDSF" is treated as a dummy variable with a value of 0 or 1 (0 for no, 1 for yes). The minimum value of the remaining variables is determined in accordance with the relevant clauses in International Building Code 2021 (IBC 2021) [56], NFPA (National fire protection association) [40], and China's Code for fire protection design of buildings [57] as the benchmark, while the maximum value of the variables is determined by increasing three streams of flow per stream of 0.55 m. The range of values for each variable is shown in Table 6.

**Table 6.** The setting of the variables' value range.

| Serial Number | Variables | Value Range |
|:---:|:---:|:---:|
| X1 | SFW  | [1200, 2850] |
| X2 | SDW  | [900, 2550] |
| X3 | CW  | [1300, 2950] |
| X4 | RDW  | [900, 2550] |

**Table 6.** *Cont.*

| Serial Number | Variables | Value Range |
|:---:|:---:|:---:|
| | LDSF | |
| X5 | 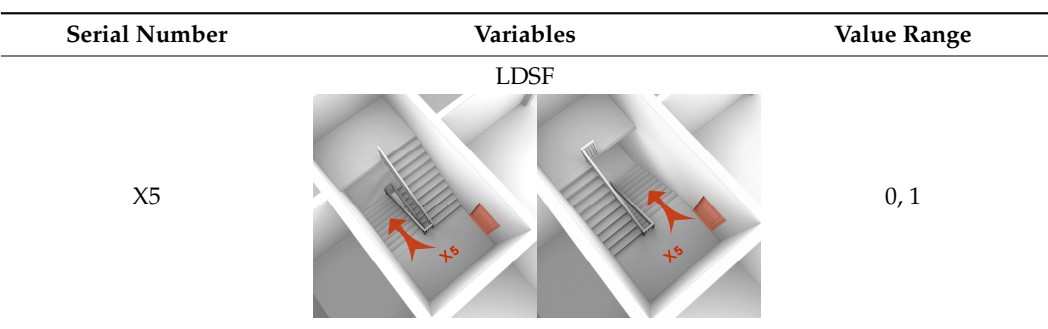 | 0, 1 |

*2.4. Data Deployment and Preference of Machine Learning Algorithms*

2.4.1. Preferential Selection of Algorithms

The evacuation procedure was simulated using the Pathfinder program. In each experiment, only a single variable was changed to acquire a total of 220 single-factor evacuation time sample data. The sample data were then split into a training set and a test set in a 9:1 ratio, and the training set data were used to create regression models using 12 widely used machine learning algorithms [11,13,29] (Table 4). $R^2$, MSE (Mean Square Error), RMSE (Root Mean Square Error), MAE (Mean Absolute Error), and MAPE (Mean Absolute Percentage Error) are used as assessment indices for the accuracy of the algorithm to predict the test set data. The formula is shown in Equations (11)–(15) [58]:

$$R^2 = 1 - \frac{\sum_{i=1}^{N}(x_i - y_i)^2}{\sum_{i=1}^{N}(x_i - \overline{x})^2} \tag{12}$$

$$MSE = \frac{1}{N}\sum_{i=1}^{N}(x_i - y_i)^2 \tag{13}$$

$$RMSE = \sqrt{\frac{1}{N}\sum_{i=1}^{N}(x_i - y_i)^2} \tag{14}$$

$$MAE = \frac{1}{N}\sum_{i=1}^{N}|x_i - y_i| \tag{15}$$

$$MAPE = 100 \cdot \frac{1}{N}\sum_{i=1}^{N}\frac{|x_i - y_i|}{x_i} \tag{16}$$

where $x_I$ refers to the ith expected output, *I* refers to the ith predicted output, $\overline{X}$ is the average of all expected outputs, N refers to the number of samples in the identification set, MSE is the expected value of the square of the disparity between the predicted and actual values, RMSE is the square root of MSE, MAE reflects the real condition of the error of the predicted value, and MAPE is the deformation of MAE. The closer $R^2$ is to 1, the smaller the MSE, RMSE, MAE, and MAPE are, the more accurate the model is.

Each indicator's value corresponds to the Random Forest algorithm were the most optimal, with $R^2$ = 0.971, the closest to 1, and the lowest values of MSE, RMSE, MAE, and MAPE (Table 7), making this regression model the most accurate. As a consequence, this algorithm was the preferred one for predicting the outcomes of multi-factor combinatorial optimization (Figure 4).

**Table 7.** Accuracy analysis of algorithms.

| Serial Number | Algorithms | MSE | RMSE | MAE | $R^2$ | MAPE |
|---|---|---|---|---|---|---|
| 1 | Decision tree | 96.009 | 9.798 | 7.466 | 0.942 | 1.704 |
| 2 | Random Forest | 46.665 | 6.831 | 5.824 | 0.971 | 1.409 |
| 3 | Adaboost | 108.014 | 10.393 | 7.709 | 0.934 | 1.775 |
| 4 | Gradient Boosting Decision Tree (GBDT) | 85.019 | 9.220 | 7.461 | 0.948 | 1.759 |
| 5 | Extra Trees | 64.128 | 8.008 | 6.583 | 0.961 | 1.541 |
| 6 | CatBoost | 99.568 | 9.978 | 7.887 | 0.939 | 1.815 |
| 7 | K-Nearest Neighbor (KNN) | 67.905 | 8.240 | 6.643 | 0.959 | 1.557 |
| 8 | Back-Propagation (BP) neural network | 771.358 | 27.773 | 19.346 | 0.534 | 4.625 |
| 9 | Support Vector Machine (SVR) | 1364.423 | 36.938 | 33.006 | 0.177 | 7.960 |
| 10 | XGBoost | 84.082 | 9.169 | 8.035 | 0.949 | 1.887 |
| 11 | Light Gradient Boosting Machine (LightGBM) | 649.480 | 25.484 | 14.749 | 0.608 | 3.379 |
| 12 | Linear Regression (Gradient Descent) | 775.047 | 27.839 | 19.234 | 0.532 | 4.641 |

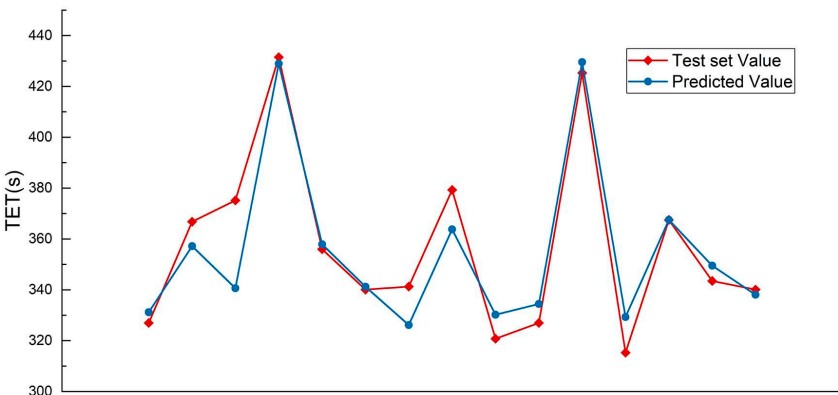

**Figure 4.** Comparison of predicted and test set values of Random Forest algorithm.

2.4.2. Parameter Combinations and Results Prediction

The goal of Monte Carlo (MC) simulation, an analytical technique for complicated systems, is to acquire predicted values through repeated random sampling of numerous random input parameters that follow a normal distribution [59]. In the physical world, there are two highly common and significant forms of stochastic processes: Bernoulli and Poisson processes, and the contrary, which states that future events will depend on previous events and can, to some extent, predict the future. There are two types of state transformation processes with time: discrete-time Markov chains and continuous-time Markov chains. The formula is as follows:

$$P(X_0 = i_0, X_1 = i_1, \ldots, X_n = i_n) = P(X_0 = i_0)p_{i_0 i_1} p_{i_1 i_2} \cdots p_{i_{0n-1} i_n} \tag{17}$$

where $X_0, X_1, \ldots X_n$ is a random sequence of variables, $i_0, i_1 \ldots i_n$ is a sequence of state. After determining the value of $P(X_0 = i_0)$, the probability of the complete path can be estimated.

In this paper, we conducted Monte Carlo simulation using the MATLAB software to create 4000 multifactor parameter combinations by picking variable values at random from each of the four variable sampling ranges, and merging them with the dummy variable. The generated data were then imported into the filtered Random Forest regression model to forecast evacuation times in the context of multi-factor interactions.

*2.5. Statistical Method*

By computing the fitted linear equation, multiple linear regression (MLR) illustrates how the dependent variable varies with the independent variable [60]. The formula is as follows [61]:

$$y_i = \delta_0 + \delta_1 x_1 + \ldots + \delta_n x_n + \gamma \qquad (18)$$

where $y_i$ is the expected value of the dependent variable, $\delta_0$ is the intercept of the *y*-axis, $\delta_1$ ($\delta_n$) is the regression coefficient of the independent variable $x_1$ ($x_n$), and $\gamma$ is the regression model's error value.

In order to conduct multiple regression analysis on the sampling data, we made use of IBM SPSS Statistics 27 software. The *p*-value in the software serves as an evaluation indicator of the significance of the component. If it is less than 0.05, it suggests a significant correlation between the independent and dependent variables, and vice versa.

In terms of the combination effect, the parameter combinations were screened by whether the safe evacuation conditions were met, after which the variable taking features were compared. The combination of safety evacuation criteria that achieved the shortest TET was further chosen, and the optimal set of solutions for each variable parameter was examined. After that, the effect on TET was analyzed for sensitive factors when they took higher and lower values.

## 3. Results and Discussion

*3.1. Results of the Simulation*

3.1.1. Determining ASET: Simulating Fire Scenarios with Pyrosim

Due to the frequent opening and closing of the staircase doors, smoke is more likely to enter a stairway when someone is moving about inside of it [62]. The high smoke concentrations tend to make it more challenging to escape, which in turn negatively affects evacuation. Hence, the critical danger time presented by the detectors installed in each of the four stairwells was utilized as the basis for calculating ASET. At the same time, slices were set up parallel (S1) and perpendicular (S2) to the horizontal plane, respectively, to visualize changes in temperature, CO concentration, and visibility during the fire. The following equation is used to determine the S1 and the four detector heights in relation to the smoke layer height [4,43]:

$$H_s \geq H_c = H_p + 0.1 H_b \qquad (19)$$

where $H_s$ is the clear height (m), $H_c$ is the danger critical height (m), $H_p$ is the average height of staff (m), and $H_b$ is the internal height of the building (m). $H_p$ is assumed to be 1.6 m, $H_b$ is taken as 3.9 m, and the height of the smoke layer is calculated as 2 m.

Therefore, the S1 and 4 detector heights were set to 2 m, i.e., Z = 4.2 + 2 = 6.2 m (at a height of 2 m on the second floor). In order to explore the temperature change in the hallway during the fire, a temperature slice (S2) was also set up in the center of the corridor next to the fire chamber.

1. Temperature analysis

As can be seen from the S1, the temperature in the fire chamber rose quickly throughout the simulation, hitting a maximum of 60 °C in 75.4 s and continuing to rise to 170 °C (Figure 5a,b). All other rooms, with the exception of the one next to the substation room, were at a safe temperature (Figure 5b). According to the profile of the detector, the temperature change trends in the four stairwells were quite similar, showing a gradual change in temperature for the first 200 s before a rapid rise. However, none of them rose above 60 °C (Figure 5c). The highest temperature during the 600 s simulation was approximately 57 °C in staircase 4, 36 °C in stairwells 1 and 2, and approximately 53 °C in stairwell 3. Due to the obstruction of the floor slab, the temperature effect range in the vertical direction was tiny from S2 placed in the center of the corridor, and the temperature progressively dropped along the horizontal direction (Figure 5d). With a floor space of 1707 m$^2$, the electrical

substation room is situated in the northwest quadrant of the floor plan. It has a high fire resistance and is removed from other principal usage rooms and the four stairwells. As a result, other areas, with the exception of the fire room and the nearby region, were not significantly affected by the temperature. The key element influencing evacuation is not temperature.

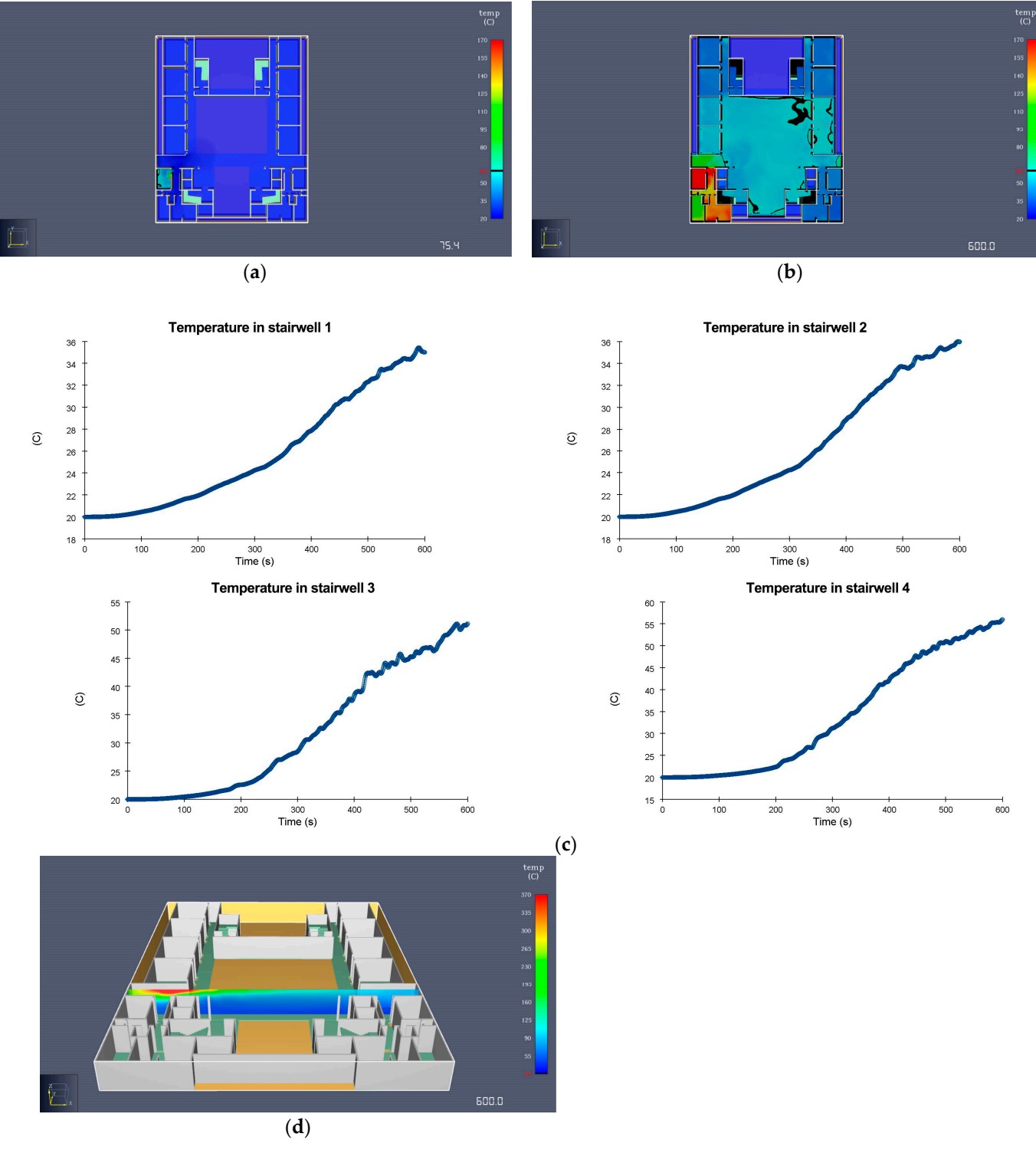

**Figure 5.** Temperature analysis: (**a**) slices of room temperature at 75.4 s; (**b**) slices of room temperature at 600 s; (**c**) curves of temperature change in four stairwells; and (**d**) slices of temperature in the *Y*-axis direction in the middle of the corridor.

2. CO concentration analysis

According to the S1, the fire room's critical CO concentration was attained at 131.8 s, and the concentration thereafter rose to 2500 ppm (Figure 6a,b). As shown by the detector's curve, the CO concentration in the four stairwells remained largely steady over a longer period of time (Figure 6c). Due to its distance from the fire room, stairwell 1's CO content did not begin to increase until roughly 350 s had passed. At around 320 s, the CO concentration in stairwell 2 began to rise toward that of stairwell 1. Closer to the fire, stairwells 3 and 4 experienced an increase in CO concentration starting at 220 and 200 s, respectively. During the rise in CO concentration, the rate of CO growth in the four stairwells was approximately the same. Among them, staircase 1's CO concentration did not approach the critical level during the simulation duration, leaving it open for personnel evacuation. However, stairwells 2, 3, and 4 did, at around 590 s, 570 s, and 450 s, respectively, reaching the hazardous critical concentration. The findings indicated that evacuation is somewhat influenced by the CO content.

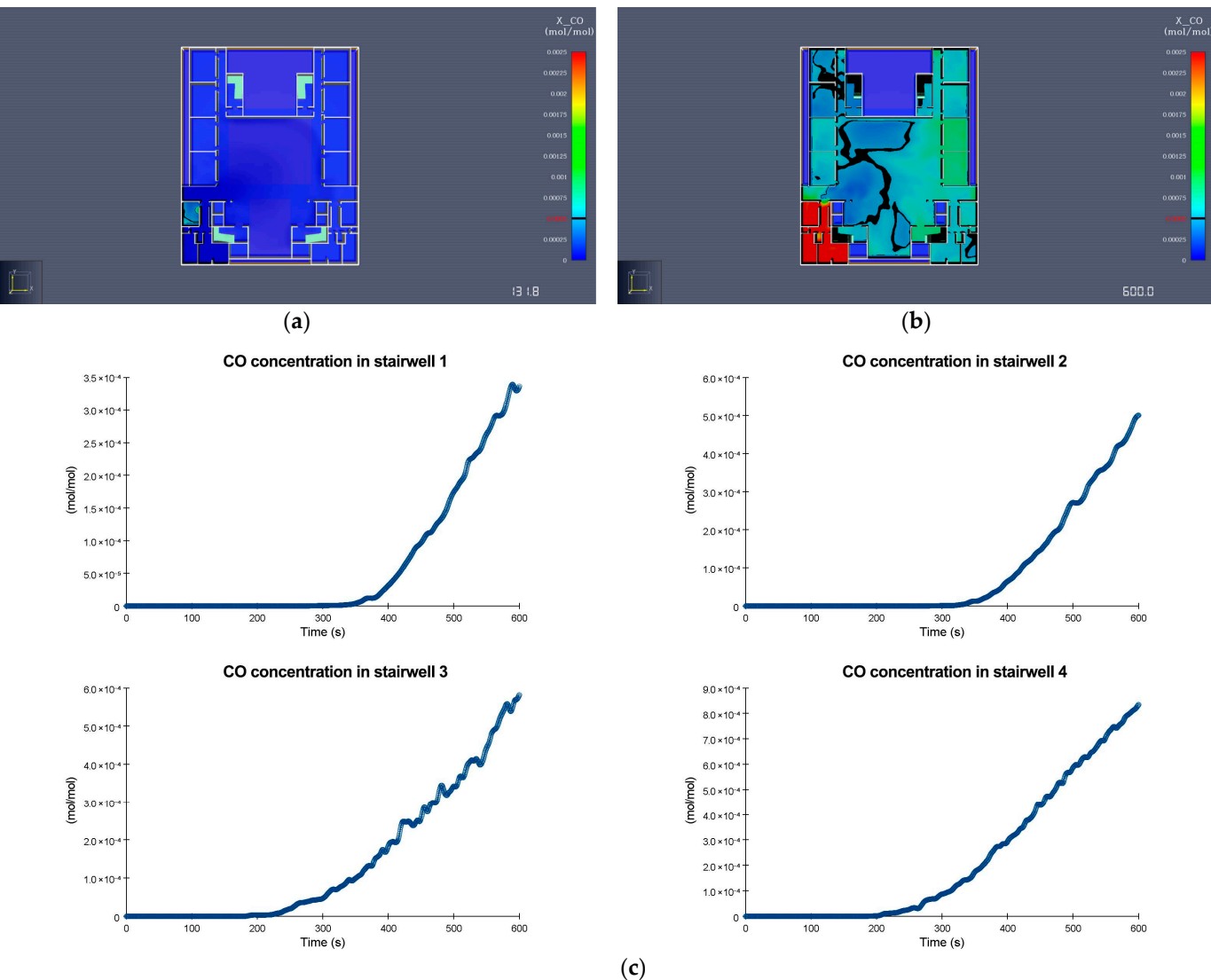

**Figure 6.** CO concentration analysis: (**a**) slice of room CO concentration at 131.8 s; (**b**) slice of room CO concentration at 600 s; (**c**) curves of CO concentration changes in four stairwells.

3. Visibility analysis

As observed in S1, the visibility in the fire room quickly decreased, hitting a critical value at 43.5 s, and then gradually dropping to 0 m (Figure 7a,b). The detector's visibility

change curve demonstrates that the visibility of the four staircases did not diminish for a while before abruptly decreasing at one point (Figure 7c). Close to the fire room, stairwells 3 and 4 saw a severe decline in visibility after 200 s and were the first to reach a critical level of risk at 250 s. At the latest at 388.6 s, stairwell 1, which is far from the burning room, reached the hazard level.

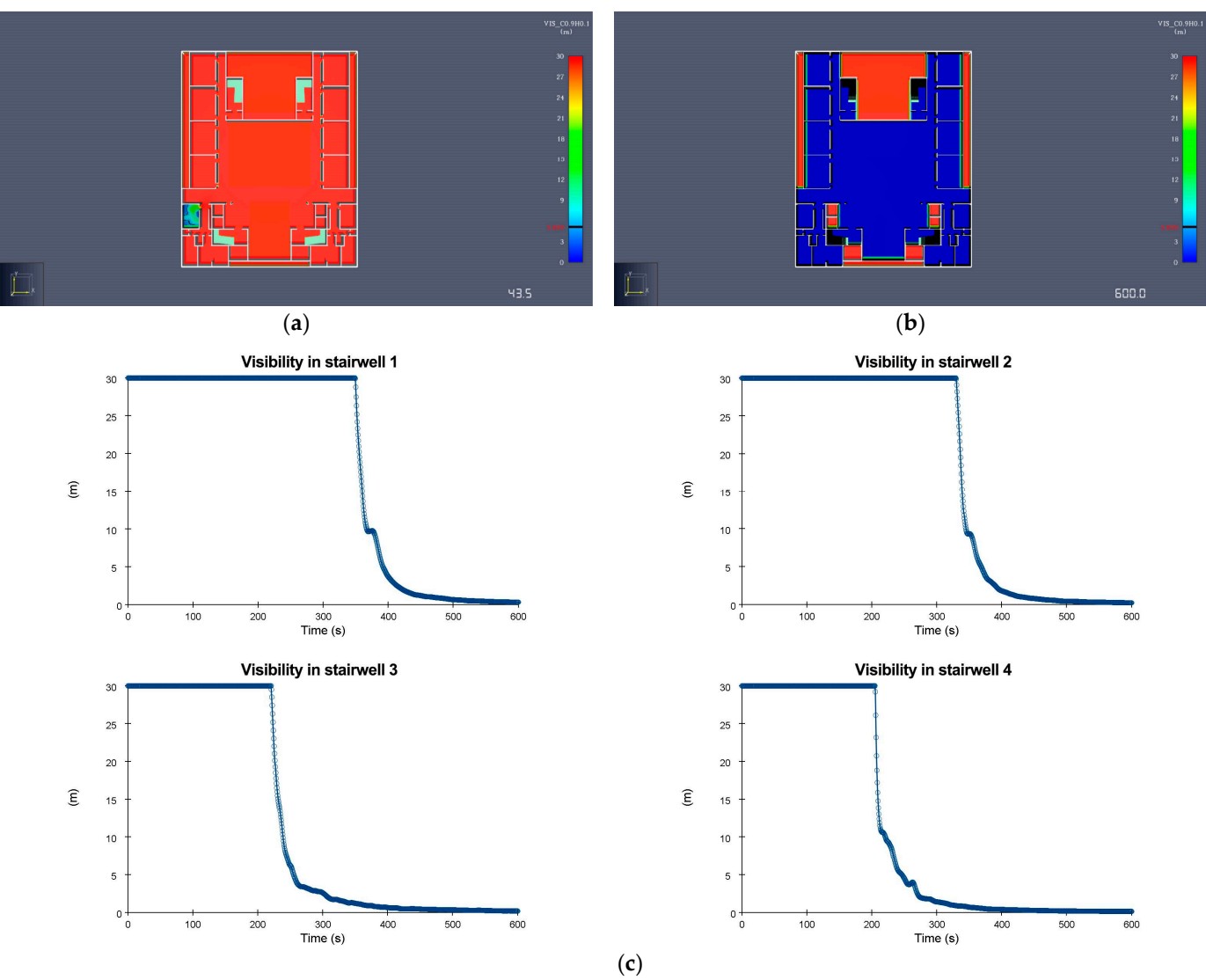

**Figure 7.** Visibility analysis: (**a**) slice of room visibility at 43.5 s; (**b**) slice of room visibility at 600 s; (**c**) curves of visibility change in four stairwells.

As can be seen from Figure 7c, the visibility inside stairwell 4 dropped to 1 m at about 300 s which was chosen as the critical time for visibility [53].

Overall, 300 s was chosen as the ASET after combining analysis of temperature (>600 s), CO concentration (>600 s), and visibility (300 s).

### 3.1.2. Determining RSET: Simulating Fire Emergency Evacuation with Pathfinder

The software's criteria for setting the age and gender distribution ratios and movement patterns of the people in the teaching building were defined based on the field research (Table 8) [39,43]. To approximate the daily utilization of the building, the average number of students was determined based on the course schedules for each classroom over the whole academic year.

**Table 8.** People parameters setting.

| Type | Gender | Ratio (%) | Shoulder Width (cm) | Height (m) | Walking Speed (m/s) |
|---|---|---|---|---|---|
| Youths | Man | 53 | 40 | 1.7 | 1.55 |
| | Woman | 37 | 37 | 1.6 | 1.5 |
| Middle-aged | Man | 5 | 41 | 1.7 | 1.52 |
| | Woman | 5 | 38 | 1.6 | 1.4 |

The building evacuation design parameters were established in accordance with the real circumstances of the teaching building (Table 9). After that, the RSET of the teaching building was determined to be 426.8 s by Pathfinder 2021 software (Figure 8).

**Table 9.** Establishment of baseline model parameters.

| Type | Parameter Setting |
|---|---|
| RDW (mm) | 1000 |
| CW (mm) | 2160 |
| SDW (mm) | 1#2# staircases 1500 |
| | 3#4# staircases 1300 |
| SFW (mm) | 1#2# staircases 1530 |
| | 3#4# staircases 1480 |
| LDSF | Away from stairwell doors |

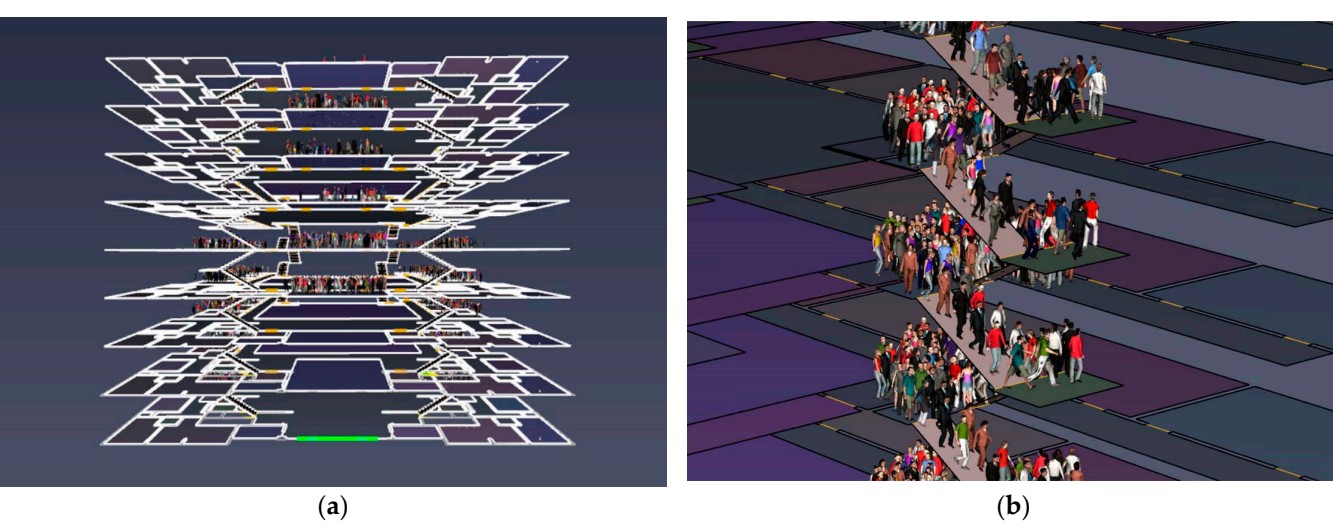

(**a**)　　　　　　　　　　　　　　　　　(**b**)

**Figure 8.** Baseline model built by Pathfinder: (**a**) Overall 3D model; (**b**) evacuation process map.

### 3.1.3. Comparison Results of ASET and RSET

According to the results of Pyrosim and Pathfinder simulations, ASET (300 s) < RSET (426.8 s), meaning that the building does not meet the standards for safe evacuation in a fire event, and would result in a 26.5% death rate (Figure 9).

### 3.2. Effect of Design Parameters on Death Rate

The key to evacuation is getting more people out in ASET, so it is imperative to first investigate how design variables affect the death rate within 300 s (ASET) [63]. Modifications to the four design parameters almost invariably result in a lower death rate than the current situation (26.5%), as depicted in Figure 10. SFW is the primary factor affecting

mortality, and increasing SFW can cause a sharp decline in death rates, with the lowest rate being 7.68% right after as SFW is 2300 mm. When the other three variables, with the exception of SFW, are altered, the death rate essentially varies above and below a particular level with a reasonably smooth pattern. Of those, SDW can contribute to an average death rate of 20.85%. Although RDW and CW have similar mortality effects, RDW is slightly superior to CW.

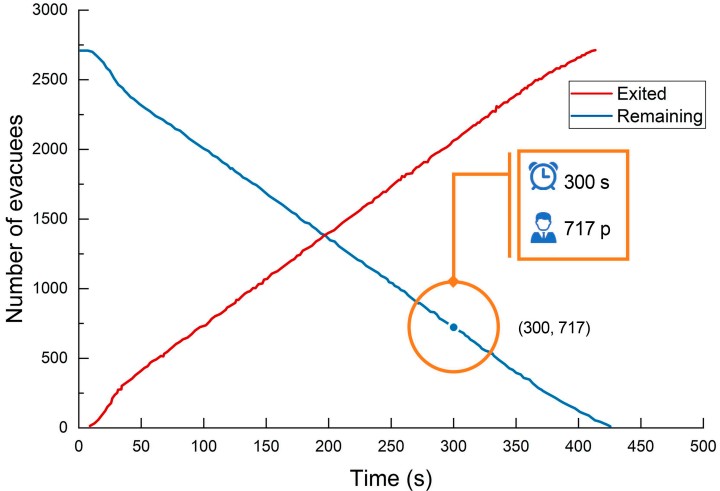

**Figure 9.** Number of evacuees.

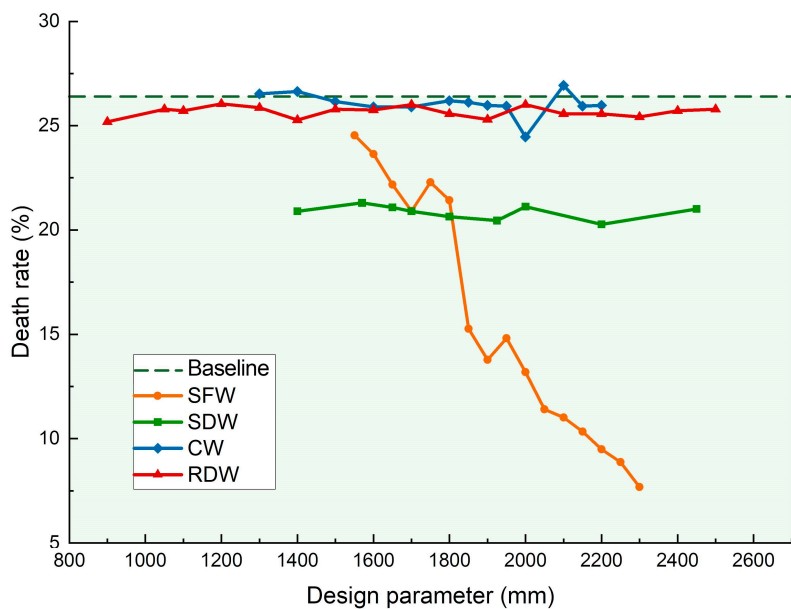

**Figure 10.** Comparison of design parameters' effect on death rate.

In other words, if only the single-factor-adjusted scenario is considered, the maximum number of people that can be accommodated in this building should not exceed 2499, otherwise fire safety evacuation cannot be achieved.

### *3.3. Data Analysis and Egress System Optimization*

#### 3.3.1. Monofactor Analysis

As indicated in Figure 11, 4000 TET data points passed the normality verification, establishing the groundwork for further data analysis.

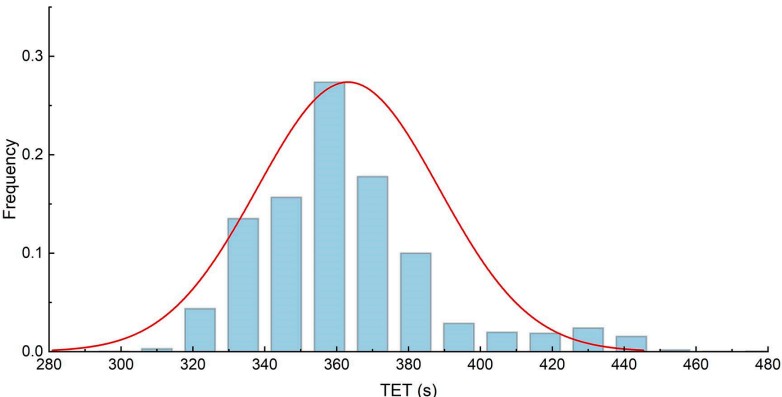

**Figure 11.** Verification of normal distribution.

The above 4000 projected data points were subjected to multiple regression analysis using the IBM SPSS Statistics 27 software. The regression model's corrected $R^2$ = 0.499 suggested that it fitted the data quite well and can partially explain the relationship between the independent and dependent variables. As shown in Table 10, the average value of the residuals is close to 0 and the standard deviation is close to 1, indicating that the data are basically normally distributed. The VIF values for each factor are between 0 and 10 (Table 11), meaning that there is no covariance in this regression model, namely, the variables are independent of one another. Moreover, Table 11 illustrates the sensitivity and magnitude of the factors' contributions to the effect of evacuation.

**Table 10.** Model residual statistics.

|  | Minimum | Maximum | Average | Standard Deviation | Number of Cases |
|---|---|---|---|---|---|
| Predicted value | 297.295 | 402.385 | 363.161 | 17.873 | 4000 |
| Residuals | −51.149 | 87.345 | 0.000 | 17.888 | 4000 |
| Standard predicted values | −3.685 | 2.195 | 0.000 | 1.000 | 4000 |
| Standard residuals | −2.858 | 4.880 | 0.000 | 0.999 | 4000 |

**Table 11.** Analysis of the variables' significance and model covariance.

| Serial Number | Variables | Beta | *p* | VIF |
|---|---|---|---|---|
| 1 | SFW | −0.560 | 0.000 | 1.052 |
| 2 | SDW | −0.444 | <0.001 | 1.017 |
| 3 | CW | −0.030 | 0.007 | 1.008 |
| 4 | RDW | 0.005 | 0.674 | 1.046 |
| 5 | LDSF | −0.089 | <0.001 | 1.030 |

1. The sensitivity factors for TET are SFW, SDW, and LDSF, all of which are design parameters of evacuation stairwells. As illustrated in Figure 12a, the stairwells are more likely to get congested than the others because they are at the end of the evacuation procedure for each level. This is in line with what the literature [55,64,65] analysis revealed.

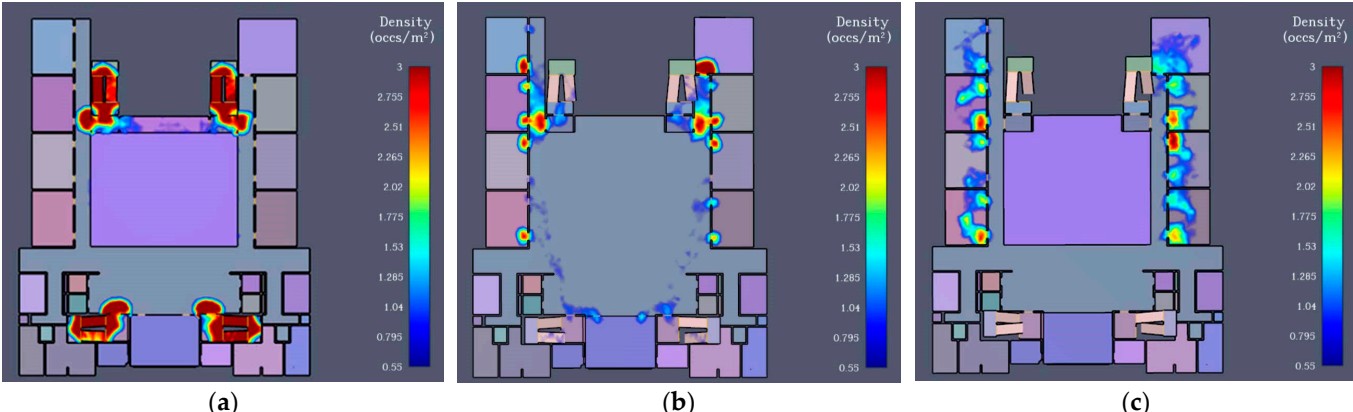

**Figure 12.** (**a**) Density distribution in the stairwell; (**b**) density distribution in the corridor; (**c**) density distribution near the room door.

2. Additionally, the results indicates that CW and RDW are not sensitive to TET, which is inconsistent with the findings of earlier investigations [66,67]. This discrepancy results from the characteristics of building planar evacuation. The layout of the building, a series of rooms clustered around an atrium, is identical to how the rooms are arranged on the side facing the outer walkway. Furthermore, the atrium does not extend all the way to the top floor, creating a bigger area to accommodate the flow of passengers on the 6th, 8th, and 9th floors, resulting in relatively mild crowding in the corridors during the evacuation process (Figure 12b). Since there is no other crowd overlay and the evacuation burden is not particularly intense, RDW primarily affects the effectiveness of early personnel evacuation from the classroom to the corridor. As a result, its impact on TET is minimal (Figure 12c).

3. SFW is the major contributor to TET (Beta = −0.560), which is due to the staircase being the final stage of evacuation and having the most people capacity. A bigger flow of people may pass through at once when the SFW is increased, which allows for a greater TET reduction [5,9,68,69].

4. SDW (Beta = −0.444) has more of an impact on TET than LDSF (Beta = −0.089). People enter the staircase by the SD and descend via the stairs. If the downstairs flight is positioned distant from the side of the stairwell door, it extends the evacuation distance for those on this floor. At the same time, they converge with the people on the upper floor. The merging behavior of the stairwell entry buffer would cut down the descending speed [38]. Meanwhile, the intensity of the behavior can be somewhat controlled by SDW, which also regulates the flow of individuals.

As shown in Figure 13, among the evacuation design factors, SFW has the greatest impact on TET. When other variables stay constant, the SFW could cut TET by 12.1%, followed by SDW, which could reduce TET by 10.1%. The reductions in TET caused by three other variables, namely SW, RDW, and LDSF, have proved steadier and more minor, with respective reductions of 5.3%, 5.6%, and 1.2%. The findings generally consistent with the results of the regression analysis shown in Table 11.

Even while no one factor alone can lower TET below ASET, SFW has the most potential. According to Figure 13a, increasing the SFW greatly reduces the TET, although there is no linear relationship between them. The trend becomes smoother after the SFW approaches 1.9 m, which indicates that there is a saturation point in the setting of SFW. When the width of the SFW reaches 3–4 streams wide (about 1.9 m), the average flow rate of people on the stairway section is up to the maximum. If the SFW is allowed to keep growing on this basis, it would lead to too many people on the stairway flight and local congestion, which would instead lengthen the TET, and even result in secondary calamities such as trampling.

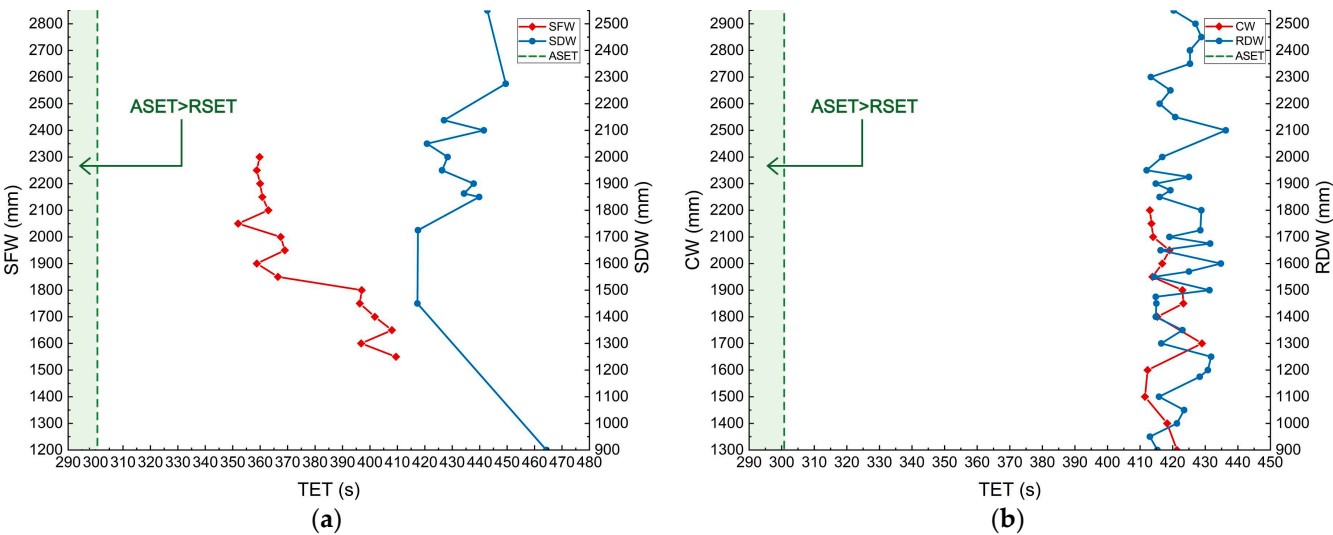

**Figure 13.** Single factor effects on TET: (**a**) SFW and SDW; (**b**) CW and RDW.

### 3.3.2. Muti-Factor Combination Effect Analysis

The first thing to observe is that, out of the 4000 combined scenarios, 3993 of them have a TET smaller than original REST (426.8 s), despite the fact that none of them have a TET smaller than ASET (300 s). The maximum improvement among all of these better-than-baseline scenarios is 29.5%, with an average optimization rate of 15.5%. For the single-factor adjusted approaches, these two numbers are 12.1% and 5.1%, respectively, indicating that the multi-factor combination is more advantageous for enhancing the evacuation performance of buildings than the single-factor solutions. The comparative data between the two are detailed in Figure 14.

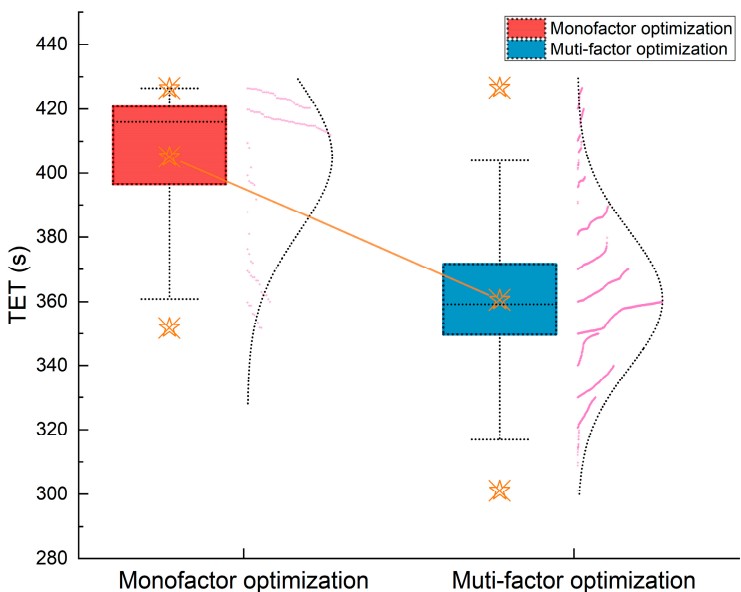

**Figure 14.** Comparison of monofactor and multi-factor optimization schemes.

For greater visualization of the parameter filtering and comparison, a parallel co-ordinate plot of the parameter combinations was drawn by Design Explorer software (Figure 15). Among them, the parameter combinations when TET is safe for evacuation are displayed in Figure 15a, and the comprehensive analysis screened by the values of SFW, SDW, and LDSF variables are shown in Figure 15b–g.

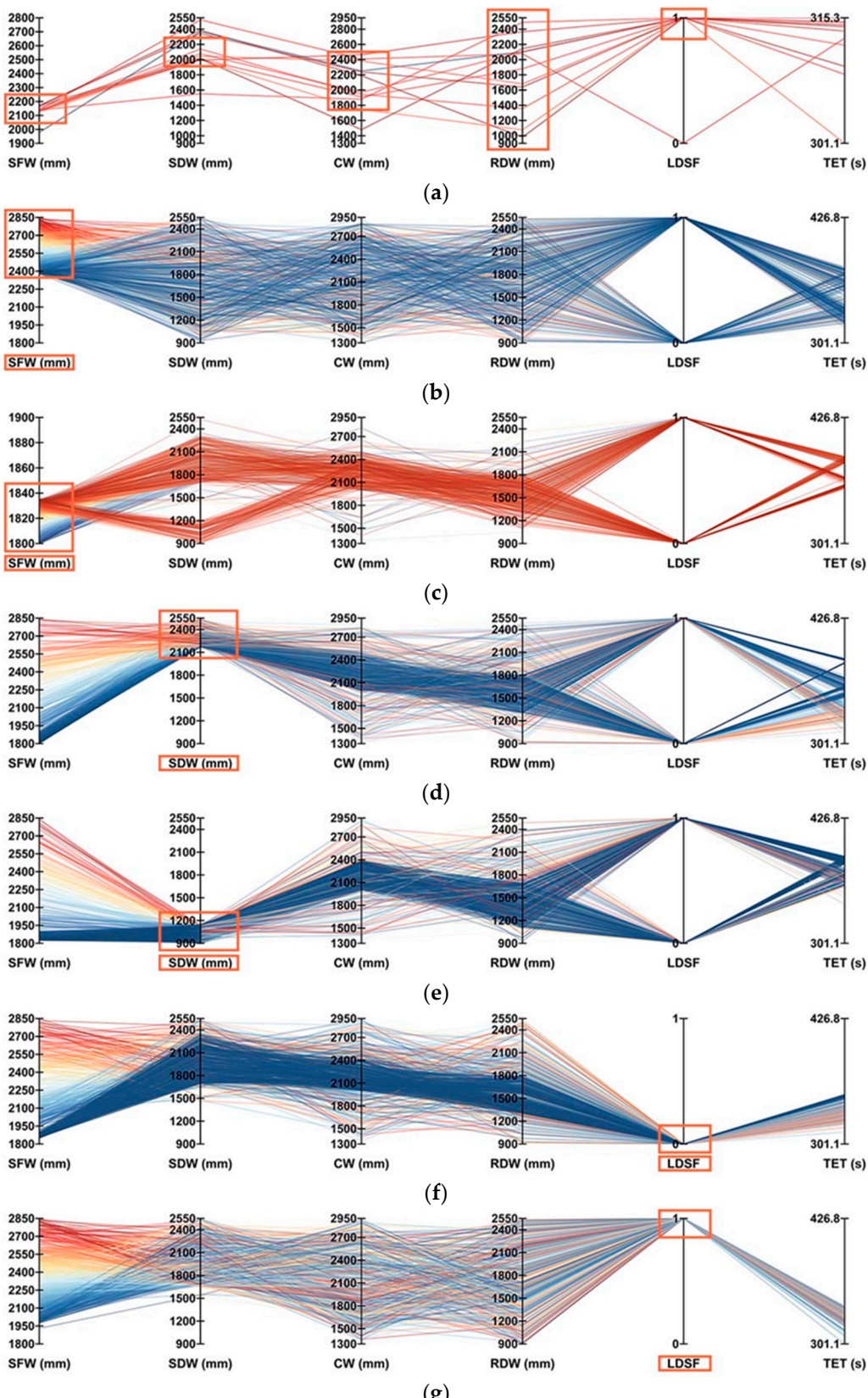

**Figure 15.** Combination effect analysis: (**a**) TET < ASET; (**b**) larger SFW; (**c**) smaller SFW; (**d**) larger SDW; (**e**) smaller SDW; (**f**) LDSF near stairwell door; (**g**) LDSF far from stairwell door. (Highlighted with orange square).

The combination of the shortest TET was then chosen to determine the properties of each factor. As can be seen in Figure 15a, SFW is concentrated in the range [2100, 2200], SDW is predominately distributed in the range [2000, 2200], CW distribution is slightly expanded in the range [1800, 2400], while RDW values do not clearly demonstrate the data aggregation phenomenon, distributing uniformly throughout the entire range of values. LDSF is almost exclusively found close to the SD side. The relevant factor's value should be the optimal set of solutions for reducing TET.

Additionally, for SFW, SDW, and LDSF, the parameter combinations corresponding to the extreme values at the two ends of the range of values were selected. It is apparent from the comparison of Figure 15b,c that raising the SFW can result in a wide distribution of TETs in the range of lower values. Regarding SDW (Figure 15d,e), widening it greatly lowers the likelihood of data occurrence in the interval of 50–70% of TET and significantly increases the density of data distribution in the interval of 0–50%. Compared to SFW and SDW, LDSF near SD lower TET less dramatically (Figure 15f,g). It agrees with the conclusion drawn from the one-way analysis above regarding the three individuals' relative contributions to TET.

If a safe evacuation by fire is to be accomplished, the maximum occupancy of this teaching building should be decreased from the existing 2707 to 2703, according to the optimal strategy in the multi-factor combination. In contrast, if the best single-factor solution is applied, the maximum number of users should be decreased to 2499. Undoubtedly, the former is more effective.

## 4. Conclusions

Fire accidents cause great damage to human beings. High-rise teaching buildings are more likely to cause casualties in the event of a fire due to the dense population inside. However, the interaction of the various components of the building egress system is not well understood, despite the fact that there are numerous articles on fire evacuation. Lack of analytical data is one of the main contributors to this issue. Therefore, after Pyrosim and Pathfinder were employed to simulate the evacuation process in a high-rise-teaching-building fire scenario, the preferred machine learning technique was utilized to increase the sample data size, and the sensitivity and contribution of the variables were examined to evaluate the combined effects of them. Through comparison studies, we found that the multi-factor optimized solutions have better compromised building evacuation performances. The main conclusions reached are as follows:

1.  Three evaluation factors—temperature, CO concentration, and visibility—are typically used to determine the ASET. The ASET in this paper was assessed through calculations and analysis to be the moment when visibility achieved its limited value in stairwell 4.
2.  The three variables relating to stairwells (SFW, SDW, LDSF) are all sensitive factors for TET, with SFW contributing the most to TET and SDW the second most. These three variables should be prioritized in the architectural program.
3.  Although neither could reach ASET, the multi-factor combinations with a maximum reduction in TET by 29.5%, outperforms the single-factor approach in terms of enhancing evacuation performance, and TET drops to the lowest when SFW [2100, 2200], SDW [2000, 2200], CW [1800, 2400], with LDSF being close to staircase door.

The findings of this paper can be applied as a design strategy for high-rise teaching buildings to mitigate fire evacuation issues, as well as to improve comprehensive plans for educational buildings that comply with fire safety design techniques, which provide certain support for improving the building egress system in both technical and theoretical investigations.

The multi-factor combination effect provides a fresh concept for future evacuation optimization design. However, there are still some limitations: the complexity of factors was not well considered in our simulated fire evacuation scenarios, and the multiplicity of actual building plan layouts was not taken into account either. These points will be covered in more detail in subsequent studies that fall outside the purview of our work.

**Author Contributions:** Conceptualization, M.Z. and B.Z.; methodology, M.Z. and B.Z.; software, M.Z., Z.Z. (Zhuo Zhang), and D.W.; validation, M.Z.; investigation, M.Z.; data curation M.Z., Z.Z. (Zuoyao Zhou), J.L. and B.L.; writing—original draft preparation, M.Z.; visualization, M.Z.; supervision, B.Z. and T.W.; project administration, B.Z. All authors have read and agreed to the published version of the manuscript.

**Funding:** This research was funded by University Student Innovation and Entrepreneurship Training Program Project of Hubei Province, China (No. S202210488075); University-Industry Cooperation Collaborative Education Program of Ministry of Education, China (No. 202102213045); and Graduate Student Quality Engineering Program of Wuhan University of Science and Technology (No. Yjg202107).

**Institutional Review Board Statement:** Not applicable.

**Informed Consent Statement:** Not applicable.

**Data Availability Statement:** Data will be made available on request.

**Conflicts of Interest:** The authors declare no conflict of interest.

## Nomenclature

| | |
|---|---|
| SFW | Stair Flight Width |
| SDW | Stairwell Door Width |
| CW | Corridor Width |
| RDW | Room Door Width |
| LDSF | Location of the Downward Stair Flight |
| TET | Total Evacuation Time |
| ASET | Available Safe Evacuation Time |
| RSET | Required Safe Evacuation Time |
| S1 | Slice 1 |
| S2 | Slice 2 |

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
