# Peer review of "Fire Egress System Optimization of High-Rise Teaching Building Based on Simulation and Machine Learning"

_fire, doi:10.3390/fire6050190_

Round 1

Reviewer 1 Report

This study used many methods, such as Pyrosim, Pathfinder, and logic analysis software, to simulate and analyze the evacuation of high-rise teaching buildings. Through simulation, ASET and RSET of buildings were compared, and various factors affecting the evacuation time of buildings were analyzed. Machine learning methods were used to study the impact of single and multiple factors on evacuation time, finally, the most sensitive factors for the evacuation time of the selected high-rise building were identified.

This paper introduces the recently well-established literature review to tell the   the need of the research and the method of used. Overall, the manuscript is complete and interesting. It is a well-done task, and it can be recommended to be published as of its form after updating some minor corrections or answer some questions as suggested below.

1.        Table 2 “ASET = RSET” represent almost safe is not suitable, as we all know ASET must have a certain margin. 

2.        There is some ambiguity in the expression of carbon monoxide concentration in Table 6.

3.        In line 458, it is said that SDW has the greatest impact on the total evacuation time, while the latter sentence also states that SDW has only 10.1% impact on the total evacuation time, which is smaller than the impact of SFW on the total evacuation time.

4.        I doubt the author's statement in line 471 that “cause the "faster is slower" phenomenon”, The effect that the flow could reduce if pedestrians push depends on the level of competitiveness but also on other important factors, like the width of the bottleneck, which is complicated.

Addressing these comments will further enhance the quality of the manuscript and its contributions.

/

Reviewer 2 Report

This paper is about the application of the concept of ASET/RSET in optimal egress analysis for a high-rise teaching building using Pyrosim for the fire simulation and Pathfinder for the evacuation analysis. The fire safety acceptance criterion in Table 2 and the ASET value used in the analysis are in question. The claimed conclusions in Section 4 have nothing to do with ASET and the related fire simulation, which is one of the major presented parts of this paper.  

Major issues:

1.      The conclusions in Section 4 have nothing to do with fire, even though fire simulation was performed for the selection of ASET in this paper.

2.      In fire safety engineering, it is necessary to ensure that ASET is greater than RSET by an acceptable margin of safety. The acceptance criterion in the analysis by the authors is simplified as ASET>=RSET (See Table 2). An insight review of ASET/RSET including the parameters affecting ASET/RSET is necessary, e.g.,

[1] Purser and McAllister, Assessment of Hazards to Occupants from Smoke, Toxic Gases, and Heat, in SFPE Handbook of Fire Protection Engineering, 2016.

3.      The ASET value of 388.6s (Line 292) is in question. This ASET value is based on the visibility (<5m) at stairwell 1. As the visibilities at stairwells 3 and 4 are less than 5 m at approximately 250s, the ASET of 388.6s is the maximum ASET in the whole building. Therefore, RSET<ASET (Table 2) is not an approximate safety criterion for the considered building with ASET=388.6, whereas it is used in the analysis in this paper. With this ASET value, the number of people who can successfully evacuate is overestimated.

4.      Without providing an acceptable margin of safety in the application of ASET/RSET as mentioned in Comment 1, the calculation of RSET should consider the impact of fire hazards, especially the impact smoke on walking speed. This is critical when the visibility is 1 m only (at around 300s in Stairwell 4 in Fig. 7) while the evacuation is being performed (See Fig. 9). In this paper, a visibility independent walking speed is applied in the calculation of RSET while the visibility is already as low as 1 m or less. Therefore, the people evacuated at the time of ASET in Fig. 9 have been overestimated. The impact of smoke on walking speed can be found in literature such as [1] in Comment 1 and [2] Jin T. and Yamada T., Irritating Effects From Fire Smoke On Visibility, Fire Science And Technology, 5:79-90. The application of ASET/RSET, and the impact of fire hazards on the progress of evacuation (RSET) has been discussed in [3] Galea E. R. et al., Coupled fire/evacuation analysis of station nightclub fire, Fire Safety Science -- Proceedings of the ninth International Symposium, IAFSS, 2008, pp. 465-476.

Other comments

5.      The sub-section Section 2.2.1 does not talk about the criterion for ASET as indicated in the heading.

6.      Instead of referencing to a paper, it is good to simply mention why the simulation duration is 600 s for the investigated fire scenario in this paper (Line 221).

7.      A mesh sensitivity study is necessary for the CFD fire simulation.

8.      The left-hand side of Equation (7) is a vector, but the right-hand side is an inner product of two vectors. Please also have a look at Equation (6).

9.      What is the ‘true value’ in Fig. 4?

10.  ‘the Z-axis and Y-axis’ (line 324) – not clear on what they are.

11.  The consequences in Table 6 should be under a certain exposure time. This exposure time is critical for the authors to select the CO threshold of 500 ppm (Line 360). This should be clearly described. 

12.  Why is one decimal included for the ASET value (388.6s) as it varies in a big range (from approximate 250s at stairwell 4 (Fig. 7))?

13.  It is not necessary to keep the decimals in the axis in Fig. 5, Fig. 7. 

14.  The limitations of the work should be discussed.

Round 2

Reviewer 2 Report

The reviewer does not agree with some of the changes to the paper and suggests a further data analysis.

1.      Line 212. ‘Following a mesh sensitivity analysis, these grids were enough [48].’ Mesh sensitivity analysis is fire scenario dependent. The number of computational cells from other irrelevant work (reference [48]) does not support the use of the number of grids in the current study. The reviewer suggests deleting this identified sentence. The reviewer would like to see that the key CFD simulation results in the paper are insensitive to the mesh used (85680 grids) by re-simulating the fire using a finer mesh.

2.      Lines 273-275. The reviewer does not understand why the RSET in the paper has been adjusted according to an irrelevant simulation. The reviewer suggests deleting this change.

3.      After the ASET is reached, no evacuation is performed in the building. It is not so important to analyze TET after the evacuation terminates due to the fire (at the ASET) in Section 3.2. This has been discussed in the cited paper [48]. Corresponding to the derived ASET value of 300 s for the given fire, the death rate is 26.5% (Line 450). Therefore, instead of the TET, it is interesting to see how the design parameters, RDW, CW, SDW etc. affect the death rate, based on the ASET of 300 s. The reviewer suggests inserting a new Section 3.2 for this analysis while the current Section 3.2 is the new Section 3.3.

Other comments:

4.      Please check the exposure times in Table 4. The exposure time corresponding to the CO concentration of 1000-2000 is longer than that for some low levels of CO.

5.      Please provide the unit for CO in Table 4.

6.      Following the change in Lines 426-429, please update the critical visibility setting in Table 5.
